# Cerebral Blood Flow in Alzheimer’s Disease: A Meta-Analysis on Transcranial Doppler Investigations

**DOI:** 10.3390/geriatrics9030058

**Published:** 2024-05-04

**Authors:** Marco Zuin, Alessandro De Vito, Tommaso Romagnoli, Michele Polastri, Eleonora Capatti, Cristiano Azzini, Gloria Brombo, Giovanni Zuliani

**Affiliations:** 1Department of Translational Medicine, University of Ferrara, Azienda Ospedaliero-Universitaria S. Anna, 44124 Ferrara, Italy; zuinml@yahoo.it (M.Z.); tommaso.romagnoli@unife.it (T.R.); michele.polastri@unife.it (M.P.); eleonora.capatti@unife.it (E.C.); cristiano.azzini@unife.it (C.A.); gloria.brombo@unife.it (G.B.); 2Department of Neurology, Stroke Division, Azienda Ospedaliero-Universitaria S. Anna, 44124 Ferrara, Italy; alessandro.de.vito@unife.it

**Keywords:** Alzheimer’s Disease, cerebral flow, transcranial Doppler

## Abstract

Background: Cerebrovascular hemodynamic impairment has been reported in Alzheimer’s disease (AD). We performed a systematic review and meta-analysis to investigate changes in cerebral blood flow (CBF) in AD patients. Methods: Data were obtained by searching MEDLINE and Scopus for all investigations published between 1 January 2011 and 1 November 2021, comparing the cerebrovascular hemodynamic between AD patients and cognately healthy age-matched controls, using transcranial Doppler (TCD) ultrasound. Results: Twelve studies, based on 685 patients [395 with AD and 290 age-matched cognitively healthy controls, with a mean age of 71.5 and 72.1 years, respectively] were included in the analysis. A random effect model revealed that AD patients, in the proximal segments of the middle cerebral artery (MCA), have a significantly lower CBF velocity, compared to controls (MD: −7.80 cm/s, 95%CI: −10.78 to −5.13, *p* < 0.0001, I^2^ = 71.0%). Due to a significant Egger’s test (t = 3.12, *p* = 0.008), a trim-and-fill analysis was performed, confirming the difference (MD: −11.05 cm/s, 95%CI: –12.28 to −9.82, *p* < 0.0001). Meta-regression analysis demonstrated that the mean CBF at the proximal MCA was directly correlated with arterial hypertension (*p* = 0.03) and MMSE score (*p* < 0.001), but inversely correlated with age (*p* = 0.01). In AD patients, the pulsatility index was significantly higher compared to controls (MD: 0.16, 95%CI: 0.07 to 0.25, *p* < 0.0001, I^2^: 84.5%), while the breath-holding index test results were significant lower (MD: −1.72, 95%CI: −2.53 to −0.91, *p* < 0.001, I^2^: 85.4%). Conclusions: AD patients have a significant impairment in relation to their cerebrovascular perfusion, suggesting that cerebrovascular hemodynamic deterioration, evaluated using TCD, may be a useful diagnostic tool.

## 1. Introduction

Alzheimer’s disease (AD) represents the most common type of dementia, affecting >45 million people worldwide [1]. In recent years, a growing body of evidence has suggested that vascular dysfunction significantly contributes to the pathogenesis of the disease. Several studies have demonstrated a cerebral blood flow decline (CBF) in AD patients, even before Aβ accumulation, as well as a direct relationship between the impairment of cerebrovascular hemodynamics and the rate of deposition of the tau protein in neurons [2,3,4,5,6]. In current clinical practice, transcranial Doppler (TCD) ultrasound remains a useful, non-invasive, and relatively inexpensive technique for the evaluation of CBF [7]. Notably, several TCD investigations have reported a reduced mean cerebral blood flow velocity and an elevated pulsatility index (PI) in patients with mild cognitive impairment (MCI) and AD [8]. Cerebral blood flow can be non-invasively investigated using TCD, by assessing different theological parameters, including mean flow velocity (MFV), PI, and breath-holding index (BHI). 

The aim of the present study was to perform an updated systematic review and meta-analysis, based on the last ten years, to investigate the cerebrovascular hemodynamics in AD patients compared to cognitively healthy age-matched controls, evaluating different rheological parameters, including MFV, PI, and BHI, using TDC, with the effort to improve and ameliorate the non-invasive diagnosis of this disease.

## 2. Materials and Methods

### 2.1. Study Design and Eligibility Criteria

This study followed the Preferred Reporting Items for Systematic Reviews and Meta-analyses (PRISMA) reporting guidelines (Appendix A) [9]. Data were obtained by searching MEDLINE and Scopus for all investigations published between 1 January 2011 and 1 November 2021, comparing the cerebrovascular hemodynamics between AD patients and cognately healthy age-matched controls, using TCD.

### 2.2. Outcomes and Definitions

The primary outcome of the study was the comparison between the mean MVF values, measured at the proximal middle cerebral artery (MCA) using TCD, expressed in cm/s. The secondary outcome was the evaluation of PI and BHI, respectively. Specifically, the former, which represents a measure of distal flow resistance and vascular wall rigidity, was obtained by subtracting the end-diastolic velocity from the peak systolic velocity and then dividing this by the middle blow flow velocity of MCA [10]. Conversely, the latter estimated the cerebral vasomotor reactivity and was calculated by dividing the percent increase in flow velocity by the length of time (seconds) during which subjects hold their breath after a normal inspiration [11].

### 2.3. Data Extraction and Quality Assessment

The selection of studies to be included in our analysis was independently conducted by two of the authors (M.Z. and G.Z.) in a blinded fashion. Any discrepancies in study selection were resolved by consulting a third author (A.D.V or C.A.). The following MeSH terms were used for the search: “Alzheimer’s disease” AND “Transcranial Doppler” OR “cerebral blood flow”. Additionally, all references cited were reviewed to identify further studies that were not included in the abovementioned electronic databases. Studies were considered eligible if (i) they provided data regarding the MVF at proximal MCA at rest; (ii) they enrolled at least fifteen patients; (iii) they presented results as mean with relative standard deviation (SD); (iv) their patients were stratified as AD patients and controls; and (v) their patients have received a diagnosis of AD. Conversely, (i) investigations without a control group (unavailable outcomes) and/or (ii) presenting CBF data based on animal studies (unusable results); (iii) those not evaluating the CBF with TCD (irrelevant outcomes); and (iv) those not written in English were excluded from the analysis.

Data extraction was independently conducted by two of the authors (E.C. and T.R.). Discrepancies between reviewers, if any, were resolved by consensus. For all studies reviewed, we extracted, for both AD patients and controls, the number of patients enrolled; mean age; female gender; mean corrected mini-mental-state-examination (MMSE) score; criteria used for the diagnosis of AD; and the prevalence of smoking, arterial hypertension (HT), and diabetes mellitus (DM), as well as mean and SD of MVF evaluated at proximal MCA, PI, and BHI. The Newcastle–Ottawa scale (NOS) was used to evaluate the methodology quality of the eligible studies [12].

### 2.4. Data Synthesis and Analysis

Continuous variables were expressed as mean ± (SD) or as median with corresponding interquartile range [IQR], while categorical variables were expressed as counts and percentages. The difference in MVF, PI, and BHI between AD patients and controls was expressed as mean difference (MD) with the corresponding 95% confidence interval (CI), using a random effect model (DerSimonian–Laird). A value of I^2^ = 0 was considered to indicate no heterogeneity, while values of I^2^ as <25%, 25–75%, and above 75% were considered to indicate low, moderate, and high degrees of heterogeneity, respectively [13]. When significant publication bias was found, we used the trim-and-fill method to adjust our results. To evaluate publication bias, both funnel plots and Egger’s tests were computed. To further appraise the impact of potential baseline confounders, a meta-regression analysis using age; gender; MMSE; and prevalence of smoke, HT, and DM as moderator variables was performed. The meta-analysis was conducted using Comprehensive Meta-Analysis software, version 3 (Biostat, Englewood, NJ, USA).

## 3. Results

### 3.1. Search Results and Included Studies 

A total of 2866 articles were obtained with our search strategy. After excluding duplicates and preliminary screening, 733 full-text articles were assessed for eligibility and 721 studies were excluded for not meeting the inclusion criteria, leaving 12 investigations fulfilling the inclusion criteria (Figure 1) [14,15,16,17,18,19,20,21,22,23,24,25].

### 3.2. Characteristics of the Population and Quality Assessment

Overall, 685 patients [395 with AD (230 females) and 290 age-matched cognitively healthy controls (184 females), with a mean age of 71.5 and 72.1 years, respectively] were included in the analysis. The general characteristics of the studies reviewed are shown in Table 1. Despite the fact that data on cognitive function were missing in one study [24] and for one control group [14], the mean MMSE for AD patients and healthy controls were 19.8/30 and 27.1/30 (*p* < 0.001), respectively. The quality assessment showed that all studies were of moderate–high quality, according to the NOS scale.

### 3.3. Cerebral Blood Flow Velocity

A random effect model, based on all twelve studies [14,15,16,17,18,19,20,21,22,23,24,25], revealed that AD patients, in the proximal segments of MCA, have a significantly lower CBF velocity compared to cognitively healthy age-matched controls (MD: −7.80 cm/s, 95%CI: −10.78 to −5.13, *p* < 0.0001, I^2^: 71.0%) (Figure 2). Both the relative funnel plot (Appendix A) and the Egger’s tests (t = 3.12, *p* = 0.008) showed evidence of potential publication bias. Therefore, a trim-and-fill analysis was performed to explore whether the publication bias influenced the stability of the results in this meta-analysis. The updated result showed an SMD: −11.05 cm/s, 95%CI: –12.28 to −9.82, *p* < 0.0001 (three studies were trimmed). The meta-regression analysis demonstrated that the MVF in MCA was directly associated with HT (*p* = 0.03) and MMSE (*p* < 0.001), but was inversely associated when age (*p* = 0.01) was used as moderator (Table 2).

### 3.4. Pulsatility Index

Data regarding the PI index were provided for nine group of patients from seven studies [14,15,16,19,20,22,25], based on 265 AD patients and 240 healthy controls, with a mean age 75.1 and 80.9 years, respectively. A random effect model demonstrated that in AD patients, PI was significantly higher compared to controls (MD: 0.16, 95%CI: 0.07 to 0.25, *p* < 0.0001, I^2^: 84.5%) (Figure 3). The funnel plot did not show significant publication bias (Appendix A). In meta-regression analysis, also in this case, PI was directly influenced by age and female gender, while it was inversely correlated with MMSE. 

### 3.5. Breath-Holding Index Test

The data of six groups, based on four investigations, enrolling 150 AD patients (mean age 69.8 years) and 94 controls (mean age 69.4 years) reported on BHI [14,17,19,20]. A random effect model evidenced that AD patients had a significant lower BHI (MD: −1.71, 95%CI: −2.53 to −0.91, *p* < 0.001, I^2^: 85.4%) (Figure 4). The relative funnel plot is presented in Appendix A. Meta-regression analysis demonstrated that only MMSE was inversely correlated with BHI.

## 4. Discussion

The current meta-analysis confirms significant cerebrovascular hemodynamic disturbances among AD patients, characterized by reduced CBF, elevated PI, and diminished BHI compared to cognitively healthy counterparts. Intriguingly, there was a direct correlation observed between MVF in the middle cerebral artery (MCA) and corrected MMSE scores, alongside an inverse correlation with age. These findings are in accordance with previous investigations, suggesting a link between higher CBF and enhanced performance in executive functioning, attention, and memory tasks [26]. On the other hand, although CBF is physiologically altered in normal ageing [27], it could also be associated with dementia incidence; indeed, according to the cerebrovascular hypothesis, chronic cerebral hypoperfusion represents one of the possible drives of neuronal dysfunction and cell death, with consequent cognitive impairment [28]. 

The presented results also evidenced a significant increase in PI in patients with AD, evidencing a higher cerebrovascular stiffness associated with a decline in intracranial vascular compliance. To this regard, older age and female sex have been reported as independent predictors of increased cerebral pulsatility, as also evidenced by our meta-regression [29,30]. Moreover, the inverse relationship between BHI and MMSE, as has previously been demonstrated, reinforces the relationship between hypoperfusion-induced cerebral hemodynamic impairment and reduced cognitive performance [31,32].

Our updated results are in line with previous findings published by Sabayan et al., who performed a systematic review and meta-analysis on CBF in patients with AD or vascular dementia, collecting studies based on TCD assessment between 1989 and 2010. These authors evidenced a state of cerebral hypoperfusion in AD, highlighting the need for further investigating the cerebrovascular reactivity and autoregulation at a regional level [33]. Considering that TCD is a non-invasive and inexpensive technique, widely available worldwide, some authors have suggested the use of this ultrasonographic approach as the screening method of choice in predicting AD risk, as well as in differentiating dementia from healthy aging [34]. However, several intrinsic and extrinsic limitations must be overcome to use TCD in daily clinical practice. First, some patients (about 10%) have no transtemporal sonic window and a similar percentage of individuals have a low-quality bone window, especially post-menopausal women [35,36]. Of the latter, the physician’s experience and the need for good patient cooperation, especially when performing the breath-holding maneuver, represent important limitations in the application of this method on a large scale [34].

From a pathophysiological perspective, CBF, arterial pulsation, and vasomotion remain fundamental in the transport of waste metabolites out of the brain, such as for the elimination of β-amyloid. In the same manner, a CBF reduction might promote a pathway that leads to AD, through a sustained reduction in the supply of oxygen and glucose to the brain and the accumulation of beta-amyloid [37,38,39]. Moreover, atherosclerosis may be associated with a diffuse injury of the cerebral vasomotor reactivity, also considering that it shares many vascular risk factors with AD, in particular hypertension and diabetes [40]. Therefore, it is reasonably fair to ask whether an intensive treatment of concomitant cardiovascular conditions, especially those associated with a lower CBF such as atrial fibrillation, ischemic cardiomyopathy, chronic heart failure, or atherosclerosis, may reduce or slow the progression of AD [41,42,43,44]. 

This study provides important updated data for future research in relation to AD. Indeed, in these patients, an early diagnosis is mandatory to ameliorate cognitive decline. Future research investigating the optimal timepoint at which CBF abnormalities, identified using TCD, become clinically useful for the early diagnosis of AD are necessary, since pathological changes precede diagnosis by decades [45]. 

Current findings underscore the potential pathophysiological significance of cerebrovascular dysfunction in the advancement of AD. Over the past two decades, mounting evidence has highlighted the pivotal role of cardiovascular factors in both the onset and progression of AD. Consequently, modifying modifiable cardiovascular risk factors may be a valid option to prevent or postpone the onset and advancement of dementia. [46]. 

Further analyses are needed to elucidate the pathophysiological mechanisms linking vascular disease and, more specifically, cerebral perfusion with AD, to promote targeted preventive strategies to prevent dementia. 

Our study has several limitations related to the observational nature of the studies reviewed, with all inherited biases. In fact, the high heterogeneity observed, which probably depends on the participants’ inclusion criteria, as well as on the study designs, may have resulted in conclusions that are not firm. Furthermore, the presence of publication bias, despite the application of the trim-and-fill method, may have also confounded the results. Moreover, we assessed the cerebrovascular hemodynamics, considering only the MCA; despite the fact that this artery represents the main vessel responsible for the perfusion of parietotemporal areas of the brain, we cannot exclude different hemodynamic conditions/disturbances in different areas of the brain. The reviewed studies did not systematically report the concomitant presence of specific cardiovascular disease, liming the results of our meta-regressions. Additionally, the treatment for AD administered in enrolled patients was generally not reported by the original investigations, so we cannot exclude potential secondary drug-induced effects. Finally, the variability of sex discrepancy from included reports and possible sex-dependent changes in the CBF in AD patients, although not statistically significant in our results, may have partially biased our findings. 

## 5. Conclusions

Despite the acknowledged limitations, the present investigation reinforces the notable impairment of cerebrovascular function in individuals with AD. Our data propose that assessing cerebrovascular hemodynamic decline through TCD could offer diagnostic utility for AD. The question of whether the disturbances in cerebrovascular hemodynamics in AD stem from the pathology itself or are a result from it remains unanswered.

## Figures and Tables

**Figure 1 geriatrics-09-00058-f001:**
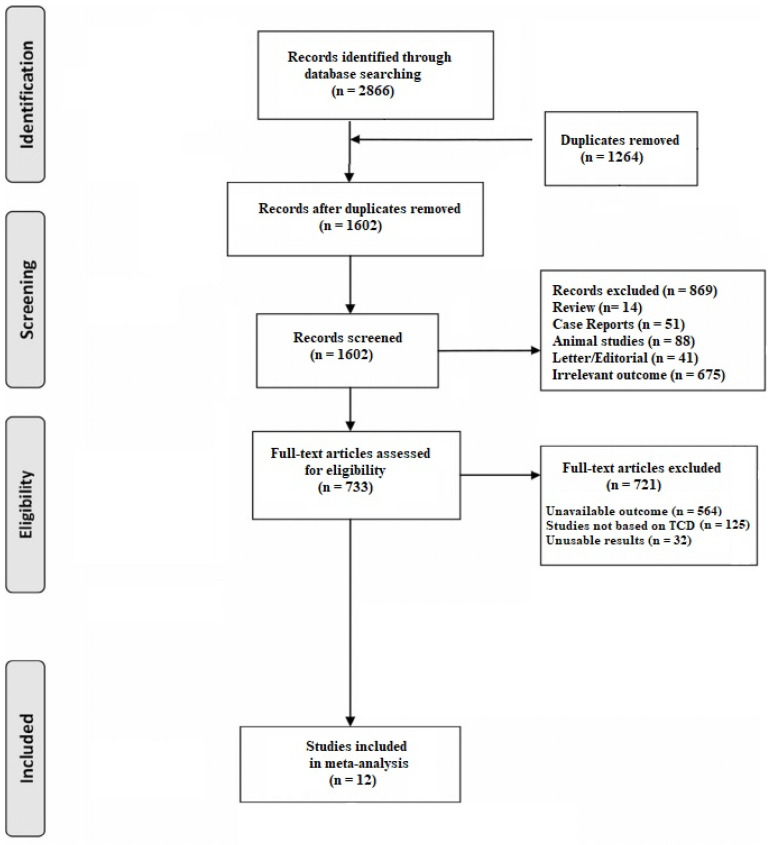
PRISMA flow chart.

**Figure 2 geriatrics-09-00058-f002:**
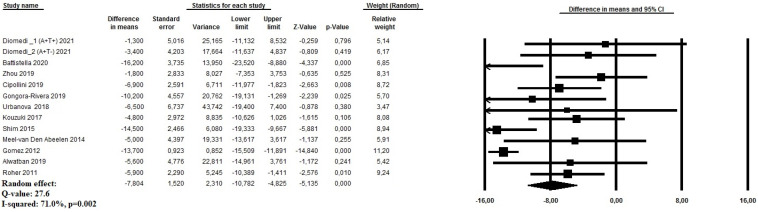
Forest plot investigating the mean difference of mean cerebral blood flow (cm/s) evaluated using transcranial Doppler ultrasound at the proximal middle cerebral artery among patients with Alzheimer’s disease and cognitively healthy age-matched controls.

**Figure 3 geriatrics-09-00058-f003:**
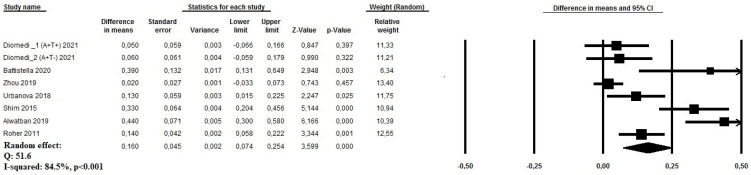
Forest plot investigating the mean difference of pulsatility index among patients with Alzheimer’s disease and cognitively healthy age-matched controls.

**Figure 4 geriatrics-09-00058-f004:**
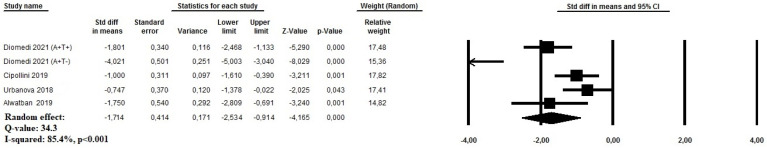
Forest plot investigating the mean difference of breath holding index tests among patients with Alzheimer’s disease and cognitively healthy age-matched controls.

**Table 1 geriatrics-09-00058-t001:** General characteristics of the population reviewed. SD: standard deviation; IQR: interquartile range; MMSE: mini-mental-state-examination; HT: arterial hypertension; DM: diabetes mellitus; NOS: Newcastle–Ottawa scale, NIA-AA; National institute on Aging and the Alzheimer’s association; NINCDS-ADRDA: National Institute of Neurological and Communicative Diseases and Stroke/Alzheimer’s Disease and Related Disorders Association; DSM: Diagnostic and Statistical Manual of Mental Disorders; IV-TR; four text revision; NR: not reported; S: single center; Retro: retrospective study. (A+T+): increase in t-tau and p-tau alongside Aβ42 decrease; (A+T−) normal levels of both t-tau and p-tau alongside Aβ42 decrease. []; Interquartile range: IQR.

Author	Year	Type	Groups	n	Female n (%)	Mean Age (SD) [IQR]	AD Criteria	MMSE (SD)	Smoke n (%)	HT n (%)	DM n (%)	NOS
Diomedi et al. [14]	2021	S; Retro	AD (A+T+)	37	21 (56.7)	71.2 (5.3)	NIA-AA	23 (3.9)	5 (13.5)	19 (51.4)	0	8
			AD (A+T−)	33	16 (48.4)	69.6 (7.2)		24 (3.2)	4 (12.1)	23 (69.7)	10 (30.3)	
			Controls	17	3 (17.6)	68.4 (8.0)	-	NR	3 (17.6)	7 (41.2)	4 (23.5)	
Battistella et al. [15]	2020	S; Retro	AD	31	27 (87.0)	79 (6.8)	NIA-AA	15 (6.4)	6 (19.0)	14 (45.0)	7 (23.0)	8
			Controls	10	8 (70.0)	74 (5.3)	-	27 (1.2)	2 (20.0)	7 (70.0)	4 (40.0)	
Zhou et al. [16]	2019	S; Retro	AD	31	16 (51.6)	69.4 (7.6)	NINCDS-ADRDA	15.8 (6.7)	NR	NR	NR	6
			Controls	30	14 (46.6)	69.6 (7.2)	-	24.7 (6.0)	NR	NR	NR	
Cipollini et al. [17]	2019	S; Retro	AD	35	10 (58.8)	72.5 (5.1)	NINCDS-ADRDA	22.6 (4.6)	5 (14.3)	13 (37.0)	2 (5.7)	8
			Controls	17	33 (94.2)	70.4 (5.6)	-	29.4 (0.6)	3 (17.6)	6 (35.3)	1 (5.9)	
Gongora-Rivera et al. [18]	2019	S; Retro	AD	26	21 (81.0)	78 (67–93)	NINCDS-ADRDA	14 (5.8)	6 (23.0)	13 (50.0)	9 (35.0)	8
			Controls	19	15 (79.0)	78 (59–90)	-	27 (3.2)	7 (37.0)	8 (42.0)	4 (21.0)	
Alwatban et al. [19]	2019	S; Retro	AD	10	6 (60.0)	68.1 (5.1)	NR	21.2 (5.9)	0	1 (10.0)	0	7
			Controls	9	5 (55.5)	71.3 (3.8)	-	29.7 (0.7)	0	5 (56.0)	2 (22.0)	
Urbanova et al. [20]	2018	S; Retro	AD	14	11 (78.5)	67.9 (11.1)	DSM-IV-TR	18 (4.6)	6 (10.2)	8 (57.1)	0	8
			Controls	24	10 (41.6)	67.8 (6.4)	-	29.1 (1.2)	7.4 (11.6)	11 (45.8)	3 (12.5)	
Kouzuki et al. [21]	2017	S; Retro	AD	42	26 (61.9)	80.5 (5.7)	DSM V	20.4 (3.6)	NR	NR	NR	6
			Controls	18	13 (72.2)	75.6 (5.5)	-	27.9 (2.4)	NR	NR	NR	
Shim et al. [22]	2015	S; Retro	AD	67	50 (74.6)	74.6 (6.2)	NINCDS-ADRDA	17.2 (4.5)	6 (3.0)	42 (21.6)	16 (8.2)	8
			Controls	52	35 (67.3)	66.2 (6.5)	-	28.2 (1.5)	2 (1.0)	24 (12.3)	8 (4.1)	
Meel-van-den Abeelen et al. [23]	2014	S; Retro	AD	12	3 (25.0)	74.0 (4.0)	NINCDS-ADRDA	22 (5.0)	NR	NR	NR	6
			Controls	24	6 (25.0)	76.0 (4.0)	-	29 (1.0)	NR	NR	NR	
Gommer et al. [24]	2012	S; Retro	AD	15	7 (46.6)	72.0 (2.0)	NINCDS-ADRDA	NR	NR	NR	NR	6
			Controls	20	10 (50.0)	70.0 (1.0)	-	NR	NR	NR	NR	
Roher et al. [25]	2011	S; Retro	AD	42	16 (38.0)	80 (6.5)	DSM V	19 (6.7)	16 (38.0)	24 (57.0)	4 (10.0)	7
			Controls	50	32 (64.0)	79 (6.4)	-	29 (1.1)	13 (26.0)	27 (55.0	3 (6.0)	

**Table 2 geriatrics-09-00058-t002:** Meta-regression analysis for the primary and secondary outcomes of the study. CI: confidence interval; MMSE: mini-mental-state-examination; HT: arterial hypertension; DM: diabetes mellitus.

Items	N° of Interactions	Coeff.	95%CI	*p*
	*Cerebral blood flow velocity*			
Age (years)	12	0.20	0.34 to 0.76	0.01
Females, (%)	12	−0.55	−0.21 to 0.11	0.52
MMSE score	12	0.72	0.32 to 1.77	<0.001
Smoking	8	0.12	−0.22 to 0.48	0.49
Hypertension	8	0.21	0.001 to 0.41	0.03
Diabetes	8	−0.23	−0.62 to 0.15	0.23
	*Pulsatility Index*			
Age (years)	7	0.006	0.01 to 0.02	0.02
Females, (%)	7	0.001	0.004 to 0.007	0.3
MMSE score	6	−0.007	−0.04 to −0.02	0.02
Smoking	7	−0.06	−0.01 to 0.009	0.68
Hypertension	6	−0.005	−0.01 to 0.03	0.87
Diabetes	6	0.01	0.004 to 0.30	0.15
	*Breath-Holding Index*			
Age (years)	6	0.002	−0.85 to 1.21	0.21
Females, (%)	6	0.02	−0.23 to 0.01	0.64
MMSE score	5	−0.09	−0.05 to −0.01	0.03
Smoking	6	0.03	−0.12 to 0.21	0.52
Hypertension	6	−0.02	−0.14 to 0.52	0.44
Diabetes	6	−0.12	−0.51 to 0.13	0.55

## Data Availability

Data will be made available on reasonable request.

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
