# Peer review of "Cerebral Blood Flow in Alzheimer’s Disease: A Meta-Analysis on Transcranial Doppler Investigations"

_geriatrics, 2024, doi:10.3390/geriatrics9030058_

Round 1
Reviewer 1 Report
Comments and Suggestions for Authors
The authors conducted the systemic review and meta-analysis about the change of cerebral blood flow (CBF) in AD patients. The manuscript reported the decrease of CBF velocity in proximal MCA, elevated pulsatility index and lowered breath holding index in AD patients comparted to normal control based on Transcranial doppler (TCD) data. The authors concluded impairment of cerebrovascular perfusion in AD patients and potential effectiveness of TCD as a useful AD diagnostic tool. The study is a well-organized and straightforward, however, there are some concerns to weaken the integrity of manuscript.
Major concerns
1. Unidentified/undefined groups: While the authors presented the groups in selected 12 studies in table 1, the meta-analysis in the cerebral blood flow velocity was performed by 15 groups including two additional groups (Urbanova-R 2018 and Urbanova-L 2018) from the study in Urbanova et al (ref 20) that did not present in Table 1. These groups were also subjected to analyze cerebral blood flow velocity (Fig 2), pulsatility index (Fig 3), and breath holding index (Fig 4). It raised serious concern about the reliability of results by lack of justification and rationale about the separating groups in the same study (even though bilateral measurement in the original article). Thus, the authors must provide the rationale or justification in the separating two groups from the study (ref 20), and update the table 1.
2. Ambiguity of excluded studies: The authors presented the number and reason of the excluded articles based the exclusion criteria which the authors set up in Fig 1. However, it is unclear what the irrelevant outcomes, unavailable outcomes, and unusable results mean, because these descriptions unmatched with the exclusion criteria which is established by the authors. Since a high number of the studies are excluded, providing the number of excluded articles which is categorized by the authors’ exclusion criteria in Fig 1 or the section 3.1 Search results and included studies will improve the rigorous of the manuscript.
3. Sex discrepancy: The studies which are subjected to analysis showed a high variation of sex distribution. Due to prominent sex-difference of CBF in the elderlies, it is hard to ignore the sex-dependent CBF change in AD patients even though the insignificant effect of sex statistically. It is highly recommended to add the limitation of the study about the variability of sex discrepancy from included reports, and possible sex-dependent change of CBF in AD patients.
Minor concerns
1. Add the glossaries: It is highly recommended to add the glossaries. It will help to avoid the confusion that is caused by use of abbreviation (ie, MVF) and authors’ term such as Urbanova-R or Groups A+T- / A+T+.
2. Typo: Check the typo (ie. Zhou et ak [16] in table 1)
Author Response
The authors conducted the systemic review and meta-analysis about the change of cerebral blood flow (CBF) in AD patients. The manuscript reported the decrease of CBF velocity in proximal MCA, elevated pulsatility index and lowered breath holding index in AD patients comparted to normal control based on Transcranial doppler (TCD) data. The authors concluded impairment of cerebrovascular perfusion in AD patients and potential effectiveness of TCD as a useful AD diagnostic tool. The study is a well-organized and straightforward, however, there are some concerns to weaken the integrity of manuscript.
Major concerns
- Unidentified/undefined groups: While the authors presented the groups in selected 12 studies in table 1, the meta-analysis in the cerebral blood flow velocity was performed by 15 groups including two additional groups (Urbanova-R 2018 and Urbanova-L 2018) from the study in Urbanova et al (ref 20) that did not present in Table 1. These groups were also subjected to analyze cerebral blood flow velocity (Fig 2), pulsatility index (Fig 3), and breath holding index (Fig 4). It raised serious concern about the reliability of results by lack of justification and rationale about the separating groups in the same study (even though bilateral measurement in the original article). Thus, the authors must provide the rationale or justification in the separating two groups from the study (ref 20), and update the table 1.
Answer: We thank the Reviewer for the comments. Considering that in the original paper publishe by Urbanova et al, the baseline characteristics were not presented according to the right and left carotid assessment, we revised the pooled analyses presenting (Figures 2,3 and 4), poling only the mean data derived from the entire cohort, as done for the other investigations. We agree with the Reviewers regarding the absence of a rationale on presenting the data as two groups. The novel pooled analysis confirmed previous results, with only non-significant changes in the third decimals.
- Ambiguity of excluded studies: The authors presented the number and reason of the excluded articles based the exclusion criteria which the authors set up in Fig 1. However, it is unclear what the irrelevant outcomes, unavailable outcomes, and unusable results mean, because these descriptions unmatched with the exclusion criteria which is established by the authors. Since a high number of the studies are excluded, providing the number of excluded articles which is categorized by the authors’ exclusion criteria in Fig 1 or the section 3.1 Search results and included studies will improve the rigorous of the manuscript.
Answer: We thank the Reviewer for the thoughtful comment. In this revised version, we have more clearly defined such definitions into the methods-exclusion criteria.
“Conversely, were excluded from the analysis those investigations i) no presenting a con-trol group (unavailable outcomes) and/or ii) presenting CBF data based on animal stud-ies(unusable results) iii) not evaluating the CBF with TCD (irrelevant outcomes) and iv) not written in English language.”
- Sex discrepancy: The studies which are subjected to analysis showed a high variation of sex distribution. Due to prominent sex-difference of CBF in the elderlies, it is hard to ignore the sex-dependent CBF change in AD patients even though the insignificant effect of sex statistically. It is highly recommended to add the limitation of the study about the variability of sex discrepancy from included reports, and possible sex-dependent change of CBF in AD patients.
Answer: We thank the Reviewer for the comment. We perfectly agree with the observation. In this revised version of the paper we have highlighted this issue into the limitations.
“Finally, although our results did not show any statistically difference in the sex-related CBF, some previous analysis, the intrinsic sex variability may have partially biased our findings. “
Minor concerns
- Add the glossaries: It is highly recommended to add the glossaries. It will help to avoid the confusion that is caused by use of abbreviation (ie, MVF) and authors’ term such as Urbanova-R or Groups A+T- / A+T+.
Answer: We thank the Reviewer for the comment. A section containing the abbreviations used in the manuscript has been added after the abstract.
- Typo: Check the typo (ie. Zhou et ak [16] in table 1).
Answer: Revised.
Reviewer 2 Report
Comments and Suggestions for Authors
In this study, the authors conducted a systematic review and meta-analysis to investigate changes in cerebral blood flow (CBF) among Alzheimer's disease (AD) patients compared to cognitively healthy age-matched controls using transcranial Doppler (TCD). They analyzed twelve studies involving 685 patients (395 with AD and 290 controls). The results indicated that AD patients exhibited significantly lower CBF velocity in the proximal segments of the middle cerebral artery (MCA) compared to controls. Furthermore, meta-regression analysis revealed correlations between CBF and factors such as arterial hypertension, MMSE score, and age. Additionally, AD patients demonstrated a higher pulsatility index and lower breath-holding index compared to controls.
These findings suggest that assessing cerebrovascular hemodynamic impairment via TCD could serve as a valuable diagnostic tool for AD. The study's methodology and presentation are commendable, warranting no concerns regarding its publication in its current form. A minor revision in English spelling is needed.
Only few comments:
Regarding the impact of these findings on AD research, they shed light on the potential role of cerebrovascular dysfunction in the pathophysiology of AD. Understanding these mechanisms could lead to the development of novel diagnostic and therapeutic approaches targeting cerebrovascular health. From my perspective, these findings support the importance of considering cerebrovascular factors in AD management and may prompt further investigation into preventive strategies targeting vascular health, such as lifestyle modifications and pharmacological interventions (please make a stronger discussion about this point). Additionally, the study highlights the need for future research to elucidate the underlying mechanisms linking cerebrovascular dysfunction and AD pathology. Please clarify whether the MMSE score is uncorrected or not. Minor editing in English spelling is required.
Comments on the Quality of English LanguageMinor editing in English spelling is required.
Author Response
In this study, the authors conducted a systematic review and meta-analysis to investigate changes in cerebral blood flow (CBF) among Alzheimer's disease (AD) patients compared to cognitively healthy age-matched controls using transcranial Doppler (TCD). They analyzed twelve studies involving 685 patients (395 with AD and 290 controls). The results indicated that AD patients exhibited significantly lower CBF velocity in the proximal segments of the middle cerebral artery (MCA) compared to controls. Furthermore, meta-regression analysis revealed correlations between CBF and factors such as arterial hypertension, MMSE score, and age. Additionally, AD patients demonstrated a higher pulsatility index and lower breath-holding index compared to controls.
These findings suggest that assessing cerebrovascular hemodynamic impairment via TCD could serve as a valuable diagnostic tool for AD. The study's methodology and presentation are commendable, warranting no concerns regarding its publication in its current form. A minor revision in English spelling is needed.
Answer: We thank the Revier for the suggestions. We have revised the entire paper for English language.
Only few comments:
Regarding the impact of these findings on AD research, they shed light on the potential role of cerebrovascular dysfunction in the pathophysiology of AD. Understanding these mechanisms could lead to the development of novel diagnostic and therapeutic approaches targeting cerebrovascular health. From my perspective, these findings support the importance of considering cerebrovascular factors in AD management and may prompt further investigation into preventive strategies targeting vascular health, such as lifestyle modifications and pharmacological interventions (please make a stronger discussion about this point). Additionally, the study highlights the need for future research to elucidate the underlying mechanisms linking cerebrovascular dysfunction and AD pathology. Please clarify whether the MMSE score is uncorrected or not. Minor editing in English spelling is required.
Answer: We have clarified that MMSE was corrected; the discussion has been modified according to the provided suggestions. We have revised the entire paper for English language.
“Current findings underscore the potential pathophysiological significance of cere-brovascular dysfunction in the advancement of AD. Over the past two decades, mounting evidence has highlighted the pivotal role of cardiovascular factors in both the onset and progression of AD. Consequently, modifying modifiable cardiovascular risk factors may be a valid option to prevent or postpone the onset and advancement of dementia. [46]. Further analyses are needed to elucidate the pathophysiological mechanisms linking vascular disease and more specifically cerebral perfusion with AD to promote target pre-ventive strategies to prevent dementia.”
Reviewer 3 Report
Comments and Suggestions for Authors
This paper is a meta-analysis of studies assessing cerebral blood flow in TCD in Alzheimer's dementia. Although reduced cerebral blood flow and increased PI levels in AD compared to controls have been described for some time, the meta-analysis provides a more evidence-based assessment and is therefore considered to be a very useful paper.
However, CBF can be calculated from TCD, but how is each calculated? Perhaps they are calculated with reference to mean flow velocity, but it would be better to describe the calculation method and formula, or whether MFV was used as a substitute. Or are you using MFV as CBF, as TCD is an assessment of velocity in the first place, which is a slightly different concept from CBF. Please consider changing the description so that we can understand each.
The following is a MINOR REVISION.
There is no official name for the abbreviation MVF, which could be Mean flow velocity, but then the abbreviation would be MFV.
Author Response
This paper is a meta-analysis of studies assessing cerebral blood flow in TCD in Alzheimer's dementia. Although reduced cerebral blood flow and increased PI levels in AD compared to controls have been described for some time, the meta-analysis provides a more evidence-based assessment and is therefore considered to be a very useful paper.
However, CBF can be calculated from TCD, but how is each calculated? Perhaps they are calculated with reference to mean flow velocity, but it would be better to describe the calculation method and formula, or whether MFV was used as a substitute. Or are you using MFV as CBF, as TCD is an assessment of velocity in the first place, which is a slightly different concept from CBF. Please consider changing the description so that we can understand each.
Answer: We thank the Reviewer for the comment. In this revised version of the manuscript, we have better highlighted the aim of the paper as following:
“Cerebral blood flow can be non-invasively investigated using TCD, by assessing different theological parameters, including mean flow velocity (MFV), PI and breath-holding index (BHI). Aim of the present study was to perform an updated systematic review and me-ta-analysis, based on the last ten years, to investigate the cerebrovascular hemodynamic in AD patients compared to cognitively normal age-matched controls, evaluating different rheological parameters, including MFV, PI and BHI by TDC with the effort to improve and ameliorate the non-invasive diagnosis of this disease”.
There is no official name for the abbreviation MVF, which could be Mean flow velocity, but then the abbreviation would be MFV.
Answer: Corrected.
Round 2
Reviewer 1 Report
Comments and Suggestions for Authors
The authors fully reflected the reviewer's comments on the revised manuscript. It is almost ready to publish except for minor points.
1. Please add what A+T- or A+T+ from Diomedi et al. stands for in the legend of Table 1.
2. Confirm the newly added sentence in section 2.3 is correct. It seems to miss the subject in the sentence.
Conversely, ??? were excluded from the analysis those investigations i) no presenting a control group (unavailable outcomes) and/or ii) presenting CBF data based on animal studies(unusable results) iii) not evaluating the CBF with TCD (irrelevant outcomes) and iv) not written in English language
3. Section number 2.3 is duplicated in the manuscript.
2.3 Data extraction and quality assessment
2.3 Data synthesis and analysis
4. Do not forget to add the keywords.
Author Response
The authors fully reflected the reviewer's comments on the revised manuscript. It is almost ready to publish except for minor points.
- Please add what A+T- or A+T+ from Diomedi et al. stands for in the legend of Table 1.
Answer: We thank the Reviewer for the observation. We have modified the Table 1 legend, as suggested.
“(A+T+): increase of t-tau and p-tau alongside Aβ42 decrease; (A+T−) normal levels of both t-tau and p-tau alongside Aβ42 decrease.”
- Confirm the newly added sentence in section 2.3is correct. It seems to miss the subject in the sentence.
Conversely, ??? were excluded from the analysis those investigations i) no presenting a control group (unavailable outcomes) and/or ii) presenting CBF data based on animal studies(unusable results) iii) not evaluating the CBF with TCD (irrelevant outcomes) and iv) not written in English language
Answer: We thank the Reviewer again for the comment. We have corrected the sentence for major clarity.
“Conversely, i) investigations without a control group (unavailable outcomes) and/or ii) presenting CBF data based on animal studies (unusable results) iii) not evaluating the CBF with TCD (irrelevant outcomes) and iv) and not written in English language were excluded from the analysis.”
- Section number 2.3 is duplicated in the manuscript.
2.3 Data extraction and quality assessment
2.3 Data synthesis and analysis
Answer: We thank the Reviewer for the observation. We have modified as suggested.
- Do not forget to add the keywords.
Answer: We thank the Reviewer for the comment. We have added keywords.